# FastGRPO: Accelerating Policy Optimization via Concurrency-aware Speculative Decoding and Online Draft Learning

**Yizhou Zhang**[1*] **Ning Lv**[1*] **Teng Wang**[2†] **Jisheng Dang**[1,3†]
[1] Lanzhou University
[2] The University of Hong Kong
[3] National University of Singapore

## Abstract

Group relative policy optimization (GRPO) has demonstrated significant potential in improving the reasoning capabilities of large language models (LLMs) via reinforcement learning. However, its practical deployment is impeded by an excessively slow training process, primarily attributed to the computationally intensive autoregressive generation of multiple responses per query, which makes the generation phase the primary performance bottleneck. Although speculative decoding presents a promising direction for acceleration, its direct application in GRPO achieves limited speedup under high-concurrency training conditions. To overcome this limitation, we propose a concurrency-aware speculative decoding framework that dynamically adjusts the drafting and verification strategy according to real-time concurrency levels, thereby maximizing the acceleration of the generation process. Furthermore, to address performance degradation arising from distributional drift between the evolving target model and the fixed draft model during training, we introduce an online draft learning mechanism that enables the draft model to continuously adapt using feedback signals from the target model. Experimental results across multiple mathematical reasoning datasets and models demonstrate that the proposed method achieves end-to-end speedups of 2.35x to 2.72x, significantly surpassing baseline approaches in efficiency. The code is available at https://github.com/yedaotian9/FastGRPO.

## 1 Introduction

Group relative policy optimization (GRPO) has recently emerged as a promising framework for enhancing the reasoning capabilities of large language models (LLMs) through reinforcement learning (Guo et al., 2025). In each training iteration, the LLM generates a group of responses to a given query. These responses are subsequently evaluated using a predefined rule-based reward function, and the resulting rewards are standardized prior to model updates via policy optimization (Shao et al., 2024). This approach exploits group-level feedback signals to guide the model toward more accurate and coherent reasoning behaviors.

However, compared to conventional supervised fine-tuning (SFT) (Ouyang et al., 2022), the GRPO training paradigm suffers from significantly lower throughput, hindering its adoption and limiting empirical exploration. This bottleneck arises primarily from the autoregressive inference required for response generation, which dominates the training pipeline. As shown in Figure 1 (a), the generation phase (i.e., response sampling) accounts for 91% to 98% of total training time across multiple mathematical reasoning datasets, making it the primary target for performance optimization.

To this end, we propose integrating *speculative decoding*, a compute-efficient and accuracy-preserving technique originally introduced by (Leviathan et al., 2023) for accelerating autoregressive inference, into the GRPO framework. Speculative decoding employs a smaller, faster "draft model"

---

[*]Equal contribution.
[†]Corresponding author.

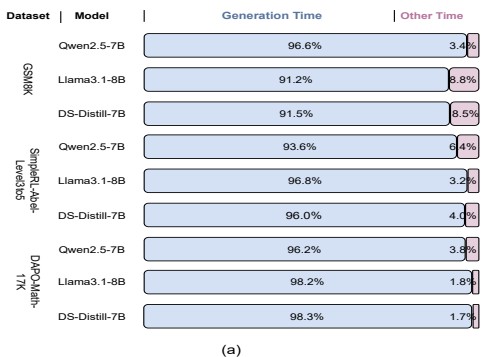 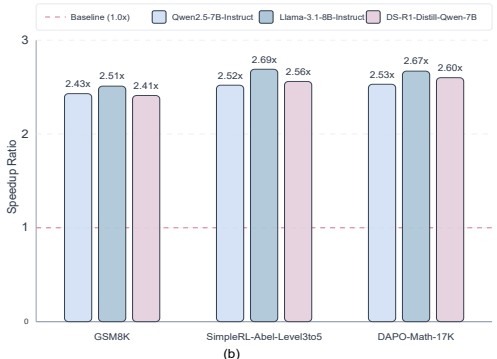

(a)                                                    (b)

Figure 1: (a) Proportion of generation time during GRPO training across multiple models and datasets, showing that generation dominates the overall time cost. (b) Speedup ratios on various models and datasets, demonstrating the significant and broad acceleration achieved by our approach.

to generate candidate token sequences in advance, which are then efficiently verified by the larger target model. This method capitalizes on the observation that autoregressive decoding is typically memory-bound rather than compute-bound, thereby enabling increased computational parallelism and improved hardware utilization. By reducing the number of direct forward passes through the target model, speculative decoding offers the potential for substantial acceleration of the *generation* phase in GRPO.

However, direct integration of speculative decoding into the GRPO generation phase yields significantly lower speedup compared to results reported in prior work (Weng et al., 2025; Li et al., 2025b), and in certain cases even leads to performance degradation. This discrepancy arises because speculative decoding is predominantly evaluated in low-concurrency, low-latency settings (e.g., edge device deployment), whereas the generation phase in GRPO operates under high-concurrency and high-latency conditions. Although speculative decoding alleviates memory bandwidth pressure, it introduces additional computational overhead. In high-concurrency scenarios, this trade-off may shift the system from a memory-bound to a compute-bound regime, thereby negating the expected performance gains.

To address this challenge, we analyze the responses generated during each step of the GRPO training process and observe significant variability in sequence lengths across batches. This variability leads to a substantial reduction in effective concurrency during the generation phase, with effective concurrency declining from a high initial batch size to nearly one as sequences terminate at different times due to uneven completion. Motivated by this observation, we propose a *concurrency-aware speculative decoding* strategy specifically designed for the generation phase in GRPO. Our method dynamically adjusts its configuration (i.e., the draft tree size and the number of verification tokens) based on real-time concurrency levels. Consequently, it achieves moderate speedup during the early, high-concurrency stage and delivers progressively greater acceleration as concurrency decreases, thereby minimizing the overall latency of the generation phase.

Furthermore, we identify a critical challenge inherent in the GRPO training framework: the target model undergoes continuous parameter updates, resulting in an increasing divergence between the target model and the fixed draft model. This distributional shift inevitably degrades model alignment, leading to declining token acceptance rates and diminishing acceleration gains over time. To mitigate this issue, we propose an *online draft learning* strategy within the GRPO loop. Our approach updates the draft model using online feedback signals derived from the evolving target model. This continual adaptation significantly enhances the draft model's representational fidelity, thereby increasing the average length of accepted speculative tokens and sustaining higher acceleration ratios throughout training, rather than allowing them to deteriorate.

We conduct experiments on Qwen2.5-7B-Instruct, Llama3.1-8B-Instruct, and other models across three mathematical reasoning datasets of varying difficulty: GSM8K (Cobbe et al., 2021), SimpleRL-Abel-Level3to5 (Zeng et al., 2025), and DAPO-Math-17K (Yu et al., 2025). As illustrated in Figure 1 (b), our proposed method substantially improves inference acceleration, achieving end-to-end speedups of 2.35x to 2.72x across different models and datasets.

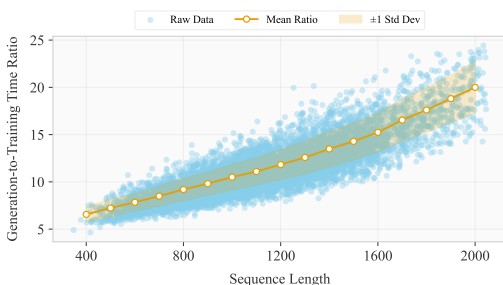 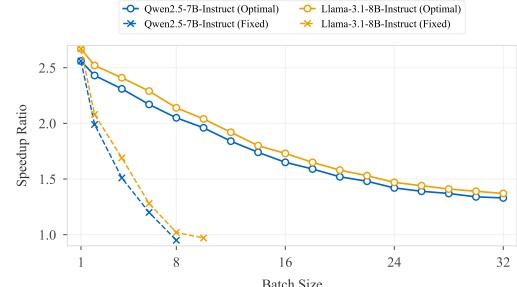

Figure 2: Ratio of generation time to parameter update time for different generation length in GRPO framework. The model is Qwen2.5-7B-Instruct and other setups are in Section 5.1.

Figure 3: Speedup Ratio vs. Batch Size under Different Models and Two Speculative Decoding Strategies. The model is Qwen2.5-7B-Instruct and other setups are in Section 5.1.

## 2 PRELIMINARIES

**Speculative Decoding.** Speculative decoding (Leviathan et al., 2023) is an inference acceleration technique for autoregressive language models that leverages a fast *draft model* to propose candidate token sequences, which are then verified in parallel by a more capable but slower *target model*. The method is motivated by two key observations: (1) not all generation steps require the full capacity of the target model; some tokens can be predicted accurately by smaller models (Hinton et al., 2015; Jaszczur et al., 2021; Hubara et al., 2018; So et al., 2021; Shazeer, 2019); and (2) in modern hardware, inference is often limited by memory bandwidth rather than compute (Vaswani et al., 2017; Park et al., 2025), leaving room for additional parallel computation. The process operates in two phases. First, the draft model autoregressively generates several candidate tokens. Then, the target model performs a single forward pass over the input prefix concatenated with these speculative tokens, using an appropriate attention mask to compute logits for all positions in parallel. Drafted tokens are only accepted if they match the target model's output. By employing strict acceptance criteria, the final generated sequences are guaranteed to be identical to those produced by the target model standalone, ensuring that the acceleration of the rollout process does not compromise the performance of GRPO.

**Draft Model Architectures.** We adopt the draft model architecture from EAGLE (Li et al., 2024b) enhanced by Feature Sampling and Partial Alignment Distillation (Gui et al., 2024). This approach trains a standalone draft model to autoregressively generate hidden states, which are then decoded into tokens using the LM head of the target LLM. The draft model takes the previously sampled tokens as input to stabilize generation, enabling effective feature-level modeling while leveraging the target model's output distribution.

**Verification Strategies.** We adopt the adaptive token tree verification strategy introduced in EAGLE-2 (Li et al., 2024a). In this approach, the draft model first expands the candidate token tree over a number of drafting steps. The candidate tokens are then reranked based on their predicted confidences. During verification, the target model evaluates all branches in parallel using tree-structured attention, and accepts the longest path whose generated tokens match the draft tokens under strict token-level comparison.

## 3 OBSERVATIONS IN GRPO TRAINING

This section empirically investigates the computational dynamics of GRPO training, identifying the generation bottleneck and intra-batch length variance. All experiments follow the setup in 5.2.

**Inference Bottleneck.** The primary time-consuming bottleneck in GRPO training is the generation phase (i.e., rollout sampling), which far exceeds the cost of parameter updates, as shown in Figure 1 (a). This disparity stems from two main factors. First, the generation cost scales linearly with the output sequence length. As shown in Figure 2, the ratio of generation to update time increases from approximately 6x to over 20x as the model matures. This aligns with prior findings

---

**Algorithm 1** FastGRPO Framework.

---

**Require:** Policy model $\pi_\theta$ (target model), draft model $\pi_d$, prompts $\mathcal{D}$, group size $G$, hyperparameter $C_{\text{peak}}$
**Ensure:** Updated $\pi_\theta$ and $\pi_d$
1: **for** iteration $= 1$ to $I$ **do**
2:      Sample a batch $\mathcal{D}_b = \{q_b\}_{b=1}^B \subset \mathcal{D}$; $\pi_\theta$ prefill $\mathcal{D}_b$; set $B_{\text{cur}} = B \times G$
3:      **while** not all sequences in $S_{\text{generated}}$ end with EOS **do**
4:          *// **Concurrency-aware Speculative Decoding***
5:          Obtain hyperparameters $L_{\text{draft}}, K_{\text{draft}}, N_{\text{verify}}$ from Eq. 3, Eq. 2, Eq. 1, based on $B_{\text{cur}}$ and $C_{\text{peak}}$
6:          **for** $\ell = 1$ to $L_{\text{draft}}$ **do**
7:              Generate children for $K_{\text{draft}}$ paths and update candidate tokens $T_{\text{candi}}$    *// draft expansion*
8:          Select $N_{\text{verify}}$ tokens from $T_{\text{candi}}$ based on their confidences    *// draft reranking*
9:          Verify selected tokens in parallel    *// verification phase*
10:         Extract accepted tokens and corresponding hidden states; update $S_{\text{generated}}$
11:         Update $B_{\text{cur}}$ and $S_{\text{generated}}$ if any sequence ends with EOS
12:      *// **Online Draft Model Updating***
13:      Draft Training phase: Update $\pi_d$ using $S_{\text{generated}}$ and corresponding hidden states
14:      Policy Training phase: Compute rewards for $S_{\text{generated}}$ and update $\pi_\theta$

---

that more capable policy models tend to generate longer, more complex outputs (Liu et al., 2025; Wu et al., 2025; Shrivastava et al., 2025). Consequently, the growing sequence length throughout training progressively inflates the inference cost. Second, the prevalence of low-reward-variance rollouts reduces data efficiency and increases the inference cost. GRPO requires response groups with non-zero reward variance to derive a learnable gradient. However, models often produce homogeneous outputs[1] where all responses are uniformly correct or incorrect, resulting in zero-variance groups that must be discarded.

**High Intra-Batch Variance in Response Lengths.** The group-based relative advantage inherently encourages length variation during response sampling. Our analysis reveals significant length variation in responses generated within a single GRPO batch. Across various models and datasets, the maximum sequence length is typically 3 to 5 times the minimum, and the range often exceeds the mean length. This heterogeneity is more pronounced on challenging datasets and correlates with intrinsic model properties like output diversity. The high intra-batch variance characteristic leads to a notable transition from high to low concurrency during the generation phase of GRPO, rather than sustaining high concurrency throughout.

## 4 FASTGRPO

**Overview.** FastGRPO integrates two mechanisms: **concurrency-aware speculative decoding** and **online draft learning**. During GRPO *generation* phase, we use speculative decoding to accelerate batched rollouts, and the decoding hyperparameters are adaptively tuned to the current concurrency (i.e., batch size). In the *parameter update* phase, we alternately freeze one model and train the other: the draft model is trained on the target model's previously generated hidden states to match its output distribution, while the target model follows standard GRPO (reward scoring and policy-gradient updates). The full procedure is summarized in Algorithm 1.

### 4.1 CONCURRENCY-AWARE SPECULATIVE DECODING FOR GRPO TRAINING

**Motivation.** Speculative decoding is primarily designed as an acceleration technique for low-concurrency inference scenarios. In principle, its performance gain arises from the verification phase: by enabling the target model to validate multiple candidate tokens in a single forward pass, it reduces the frequency of parameter transfers from GPU memory to on-chip SRAM, thereby alleviating memory bandwidth pressure. However, this benefit comes at the cost of increased computational load. Specifically, additional FLOPs are incurred during draft generation and the verification of invalid speculative tokens.

---

[1]For instance, on a moderately difficult dataset SimpleRL-Abel-Level3to5, about 20% of generated groups are filtered out. This problem intensifies on datasets that are either too simple or too complex for the model.

We study the impact of batch size on speculative decoding speedup. As shown in Figure 3, using a configuration optimized for small batches ($B = 1$) leads to significant performance degradation at larger batch sizes, with speedup dropping below 1.0x at higher concurrency, indicating growing computational overhead. In contrast, dynamically adjusting key speculative decoding parameters (e.g., draft tree size and number of verified tokens) as batch size increases effectively mitigates this degradation. The speedup remains robust under high concurrency, demonstrating that adaptive tuning is essential for maintaining efficiency. These results highlight the need to align speculative decoding hyperparameters with system concurrency to balance compute and memory demands.

Our earlier analysis reveals that the generation phase in GRPO does not operate under static concurrency; instead, it evolves dynamically from high to low concurrency as sequences complete at different times due to variable lengths. Therefore, a method that adapts speculative decoding parameters in real time according to the current effective batch size is both practical and effective. Such adaptivity enables sustained acceleration throughout the generation phase, minimizing the overall latency of the GRPO training pipeline.

**Analysis of Operational Intensity.** We begin with a theoretical analysis of operational intensity (Williams et al., 2009; Yang et al., 2020; 2021), defined as the ratio of computation (in FLOPs) to memory traffic (in bytes). This metric is central to understanding the performance characteristics of low- versus high-concurrency inference scenarios. For general matrix multiplication (GEMM) (Kurzak et al., 2010; Xu et al., 2022), the operational intensity scales as:

$$I_{\text{GEMM}} = \frac{2BD_{\text{in}}D_{\text{out}}}{(BD_{\text{in}} + D_{\text{in}}D_{\text{out}} + BD_{\text{out}}) \cdot s} = \frac{2B}{\left(\frac{B}{D_{\text{out}}} + 1 + \frac{B}{D_{\text{in}}}\right) \cdot s} \approx \frac{2B}{s},$$

where the input matrices have dimensions $B \times D_{\text{in}}$ and $D_{\text{in}} \times D_{\text{out}}$, respectively, and $s$ is the bytes per element (e.g., 2 for BF16 or FP16). The numerator represents the total number of arithmetic operations, while the denominator accounts for the memory traffic, i.e., the total storage size of the input and output matrices. The final approximation holds when $B \ll \min\{D_{\text{in}}, D_{\text{out}}\}$. In the Transformer architecture, the time cost is predominantly dominated by a series of GEMM operations. Therefore, $I_{\text{GEMM}}$ can serve as a reasonable approximation of the operational intensity of Transformer models.

Theoretically, the peak operational intensity of modern accelerators is determined by their hardware specifications, specifically by the ratio between peak FLOPS and memory bandwidth. In practice, however, analytically determining this peak value is challenging due to implementation-specific factors such as kernel optimizations, memory tiling strategies, and SRAM capacity limitations in matrix computations. These factors introduce non-ideal hardware behaviors that are difficult to model precisely. Instead, we approximate this theoretical limit through empirical profiling. We define a hardware- and architecture-dependent constant, $C_{\text{peak}}$, as the batch size $B$ at which $I_{\text{GEMM}}$ reaches its practical peak operational intensity for a given model and a specific GPU. In other words, $C_{\text{peak}}$ represents the concurrency level at which the system transitions from being memory-bound to compute-bound. (See Appendix B for the detailed measurement methodology.)

**Concurrency-aware Strategy.** We propose an adaptive speculative decoding strategy that dynamically adjusts the number of verified tokens per sequence (denoted as $N_{verify}$) and the draft tree size based on the level of concurrency. Our implementation follows EAGLE-2 (Li et al., 2024a), where the draft tree structure is governed by two key hyperparameters: $K_{\text{draft}}$, the number of candidate tokens proposed by the draft model per forward pass, and $L_{\text{draft}}$, the number of sequential drafting steps.

To make $I_{\text{GEMM}}$ approximately equal to $C_{\text{peak}}$ and thereby fully utilize both computational and memory bandwidth resources, we propose setting:

$$N_{\text{verify}} = \frac{C_{\text{peak}}}{B}. \tag{1}$$

As previously discussed, $I_{\text{GEMM}}$ is proportional to the batch size $B$, and $C_{\text{peak}}$ represents the optimal batch size at which the GPU achieves peak hardware efficiency under a given configuration. In the verification stage of speculative decoding, the effective batch size for the GEMM operations is determined by the total number of tokens verified per batch, given by the product $N_{\text{verify}} \times B$.

By maintaining $N_{\text{verify}} = C_{\text{peak}}/B$, this effective batch size remains approximately constant at $C_{\text{peak}}$, ensuring that the system operates at the compute-memory balance point. This maximizes hardware utilization and achieves the optimal speedup for speculative decoding under varying levels of concurrency.

The draft tree size should scale with $N_{\text{verify}}$, as longer verification sequences can only yield benefits if the draft model produces sufficiently diverse candidates. Therefore, we propose both $K_{\text{draft}}$ and $L_{\text{draft}}$ should be positively correlated with $N_{\text{verify}}$.

To ensure that the number of tokens generated by the draft model is both sufficient and effective, we propose this configuration for $K_{\text{draft}}$:

$$K_{\text{draft}} = \min\left(N_{\text{verify}} - 1,\ K_{\text{draft}}^{\max}\right). \tag{2}$$

Setting $K_{\text{draft}} = N_{\text{verify}} - 1$ is typically sufficient for two reasons. First, the draft model faces similar compute and memory constraints as the target model; keeping $K_{\text{draft}} \leq N_{\text{verify}}$ ensures the drafting phase remains compute-memory balanced. Second, one of the $N_{\text{verify}}$ tokens is the root of the draft tree, which is generated by the target model in the prior step and does not require speculative expansion. Thus, only $N_{\text{verify}} - 1$ slots are available for draft-generated tokens. The upper bound $K_{\text{draft}}^{\max}$ prevents inefficiency from excessive branching. The total number of candidates, $1 + K_{\text{draft}} + (L_{\text{draft}} - 1) \cdot K_{\text{draft}}^2$, often far exceeds $N_{\text{verify}}$, so large $K_{\text{draft}}$ leads to many unverified candidates and wasted computation. It also increases tree construction overhead with diminishing gains in acceptance rate. Hence, $K_{\text{draft}}^{\max}$ balances speculation coverage with efficiency.

For $L_{\text{draft}}$, we propose a design that is positively correlated with $\log_2(N_{\text{verify}})$. This is motivated by the exponential growth of the draft tree's candidate space with depth: deeper trees enable combinatorial expansion of candidate sequences, allowing longer expected acceptance length under larger verification budgets. Experimentally, we refine this principle into a practical instantiation:

$$L_{\text{draft}} = \min\left(\left\lfloor \log_2\left(\frac{N_{\text{verify}}}{\alpha}\right)\right\rfloor,\ L_{\text{draft}}^{\max}\right), \tag{3}$$

where $\alpha$ is a hyperparameter that encodes the approximation quality of the draft model. A stronger draft model (i.e., one with higher predictive accuracy) corresponds to a smaller $\alpha$, enabling deeper speculation as $N_{\text{verify}}$ increases. Conversely, weaker models require a larger $\alpha$ to prevent over-speculation and the generation of invalid branches.

This design accounts for the autoregressive nature of the draft model: prediction fidelity degrades with depth due to error accumulation, causing the likelihood of valid continuations to decay exponentially toward leaf nodes. As a result, excessive depth yields diminishing returns and may even incur negative gains from increased overhead in the drafting phase. The upper bound $L_{\text{draft}}^{\max}$ further ensures robustness and computational efficiency across diverse workloads.

### 4.2 Draft Model Learning with Online Target Feedback

In the GRPO training framework, the target model undergoes continuous policy updates, resulting in a time-varying output token distribution. When a fixed draft model is employed throughout the training process, its alignment with the target model progressively deteriorates, leading to a reduction in the average acceptance length during speculative decoding. This degradation directly translates into a declining speedup ratio, as illustrated in Figure 4.

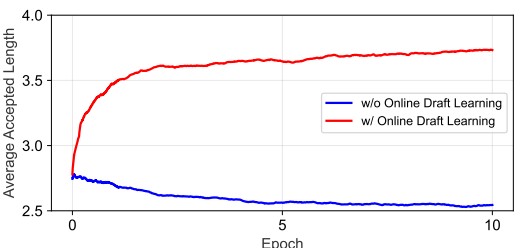

Figure 4: Average accepted length versus training steps in GRPO, with and without online draft learning. The model is Qwen2.5-7B-Instruct and the dataset is SimpleRL-Abel-Level3to5.

To mitigate this misalignment, we introduce an *online draft learning* strategy: at each GRPO iteration, the draft model is updated online using responses generated by the current target model. This enables the draft model to adapt dynamically to the evolving target policy, thereby preserving and enhancing its predictive consistency over time. As demonstrated in Figure 4, this

approach yields a steadily increasing acceptance rate, in contrast to the performance decay observed with a static draft model.

A key benefit of online draft learning arises from the quality and policy relevance of the supervision signal. Unlike conventional pre-training on static, offline datasets, the draft model learns from on-policy generations produced by the target model, which are inherently aligned with its current reasoning dynamics. This facilitates more accurate modeling of the target's output distribution. Moreover, data that cannot be used to train the target model due to non-zero reward can still serve as useful supervision signals for the draft model, thereby reducing data wastage present in the traditional GRPO framework.

Furthermore, the computational overhead of online draft learning is negligible, contributing only approximately 2% to 3% of the total training cost across various models and datasets. This efficiency is attributed to the fact that, in standard draft model training, the primary computational burden lies in executing forward passes on the target model to obtain supervisory signals (i.e., logits and hidden states). In our GRPO pipeline, these signals are naturally generated during the generation phase and can be cached without additional computation. Consequently, the draft model is trained using supervision that is effectively "free," rendering the online draft learning approach both highly effective and computationally practical.

## 5 EXPERIMENTS

### 5.1 EXPERIMENTAL SETUPS

**Models and Datasets.** We evaluate our method on Llama3.1-8B-Instruct (Dubey et al., 2024), DeepSeek-R1-Distill-Qwen-7B (Guo et al., 2025), and multiple models from the Qwen (Yang et al., 2024) series. Experiments are conducted on three mathematical reasoning datasets of increasing difficulty: GSM8K (Cobbe et al., 2021) (grade-school math), SimpleRL-Abel-Level3to5 (Zeng et al., 2025) (mid-level reasoning), and DAPO-Math-17K (Yu et al., 2025) (advanced and diverse problems). This setup enables evaluation of our method's effectiveness and generalization across model families and complexity levels.

**Pre-training of Draft Model.** Following prior work (Li et al., 2024b; Gui et al., 2024), we use ShareGPT-68K (Aeala, 2023) as the pre-training dataset for the draft model. The draft model is trained for 10 epochs with a batch size of 16. The learning rate is set to 1e-4, with the AdamW optimizer (Loshchilov & Hutter, 2017).

**Metrics.** We use **speedup ratio** (Li et al., 2024b), defined as the ratio of baseline wall-clock time to our method's time. We report: (1) **Generation speedup ratio (Gen SR)**: speedup during the generation phase, reflecting gains from speculative decoding; (2) **End-to-end speedup ratio (E2E SR)**: overall speedup across the full GRPO pipeline.

### 5.2 THE SPEEDUP OF GRPO PROCESS

**Online Draft Learning.** During the generation phase, the target model produces responses using speculative decoding. The prompt batch size per GPU is set to 4, and we sample 8 responses for each prompt. The target model is optimized with a learning rate of 1e-6, while the draft model is trained with a learning rate of 5e-5. The entire training process runs for 10 epochs using the AdamW optimizer. All experiments are conducted using GPU H800 SXM accelerators.

**Comparison.** As shown in Table 1, we use standard autoregressive decoding as the baseline, with a speedup ratio of 1.00x. We also compare against three established speculative decoding methods: EAGLE-2 (Li et al., 2024a), HASS (Zhang et al., 2024) and EAGLE-3 (Li et al., 2025b). The results demonstrate that our method significantly outperforms the unmodified speculative decoding baselines. Within the GRPO framework, our approach achieves a substantial end-to-end speedup of 2.35x to 2.72x. Notably, all experiments utilize speculative decoding with strict acceptance criteria, ensuring that the post-training model performance remains consistent with the vanilla baseline.

Table 1: Acceleration comparison in terms of Generation Speedup Ratio (Gen SR) and End-to-End Speedup Ratio (E2E SR) across different models on three mathematical reasoning datasets of increasing difficulty: GSM8K, SimpleRL-Abel-Level3to5 and DAPO-Math-17K. Q2.5-7B-I, L3.1-8B-I, DS-R1-Qwen-7B, Q2.5-Math-7B, Q2.5-Math-7B-I represent Qwen2.5-7B-Instruct, Llama3.1-8B-Instruct, DeepSeek-R1-Distill-Qwen-7B, Qwen2.5-Math-7B, and Qwen2.5-Math-7B-Instruct, respectively.

| | | GSM8K | | SimpleRL-Abel-Level3to5 | | DAPO-Math-17K | | Average | |
|---|---|---|---|---|---|---|---|---|---|
| Model | Method | Gen SR | E2E SR | Gen SR | E2E SR | Gen SR | E2E SR | Gen SR | E2E SR |
| Q2.5-7B-I | EAGLE-2 | 1.17 | 1.16 | 1.12 | 1.11 | 1.10 | 1.09 | 1.13 | 1.12 |
| | HASS | 1.21 | 1.20 | 1.15 | 1.13 | 1.11 | 1.10 | 1.16 | 1.14 |
| | EAGLE-3 | 1.28 | 1.26 | 1.22 | 1.20 | 1.14 | 1.13 | 1.21 | 1.20 |
| | FastGRPO | 2.66 | 2.43 | 2.91 | 2.52 | 2.78 | 2.53 | **2.78** | **2.49** |
| L3.1-8B-I | EAGLE-2 | 1.25 | 1.22 | 1.18 | 1.17 | 1.14 | 1.13 | 1.19 | 1.17 |
| | HASS | 1.29 | 1.26 | 1.23 | 1.22 | 1.17 | 1.16 | 1.23 | 1.21 |
| | EAGLE-3 | 1.35 | 1.31 | 1.29 | 1.28 | 1.24 | 1.23 | 1.29 | 1.27 |
| | FastGRPO | 3.04 | 2.51 | 2.92 | 2.69 | 2.83 | 2.67 | **2.93** | **2.62** |
| DS-R1-Qwen-7B | FastGRPO | 2.69 | 2.41 | 2.81 | 2.56 | 2.73 | 2.60 | **2.74** | **2.52** |
| Q2.5-Math-7B | FastGRPO | 2.75 | 2.53 | 2.96 | 2.72 | 2.84 | 2.66 | **2.85** | **2.64** |
| Q2.5-Math-7B-I | FastGRPO | 2.62 | 2.35 | 2.70 | 2.53 | 2.64 | 2.49 | **2.65** | **2.46** |

## 5.3 ABLATION STUDY

**Online Draft Learning.** We conduct experiments to evaluate the effectiveness of online draft learning in accelerating the GRPO training process, employing the Qwen2.5-7B-Instruct model. We compare Gen SR and E2E SR with and without online draft learning, under identical configurations as detailed in Section 5.2. As shown in Table 3, the proposed online draft learning strategy significantly enhances computational efficiency, increasing the generation speedup ratio by approximately 0.7x to 0.9x across various experimental settings. This improvement underscores the strong correlation between the draft model's predictive accuracy and the policy alignment of its training data. Furthermore, we observe that the training process may induce a degree of overfitting in the draft model to the current target policy. However, in the context of speculative decoding within GRPO, such overfitting is not detrimental; rather, it enhances prediction accuracy and contributes positively to the overall acceleration.

Moreover, we evaluate the performance of the draft model before and after online draft learning using SimpleRL-Abel-Level3to5 as the training dataset. The online draft learning procedure follows the setup detailed in Section 5.2. Experiments are conducted with a batch size of 1 and a sampling temperature of 0. Following established evaluation protocols (Weng et al., 2025), we assess the draft model across three

Table 2: SR and average acceptance length ($\tau$) on three benchmarks. The target model is Qwen2.5-7B-Instruct.

| | MT-bench | | HumanEval | | GSM8K | | Average | |
|---|---|---|---|---|---|---|---|---|
| State | SR | $\tau$ | SR | $\tau$ | SR | $\tau$ | SR | $\tau$ |
| Before | 1.71 | 3.34 | 2.01 | 3.87 | 2.11 | 3.82 | 1.94 | 3.68 |
| After | 1.72 | 3.37 | 2.05 | 3.93 | **2.45** | **4.55** | 2.07 | 3.95 |

distinct task categories: multi-turn dialogue, code generation, and mathematical reasoning. The corresponding benchmark datasets are MT-bench (Zheng et al., 2023), HumanEval (Chen, 2021), and GSM8K (Cobbe et al., 2021), respectively. The following metrics are used to evaluate acceleration performance: (1) **Speedup Ratio (SR)**: The measured end-to-end speedup relative to vanilla autoregressive decoding. (2) **Average Acceptance Length** ($\tau$): The average number of tokens accepted by the target model per verification step.

As shown in Table 2, the draft model after online draft learning maintains performance comparable to its pre-online-training counterpart on general-purpose tasks, indicating no degradation in generalization capability. Notably, it achieves substantial gains in-domain, where the speedup ratio increases by 0.34. These results indicate that online draft learning enhances the draft model's alignment with the target domain without compromising its versatility.

**Concurrency-aware Speculative Decoding.** We conduct experiments to evaluate the effectiveness of our proposed concurrency-aware speculative decoding. The model is Qwen2.5-7B-Instruct, and other experimental setups are identical to those described in Section 5.2. We compare our method against the following strategies: (1) Vanilla autoregressive generation with early termina-

Table 3: Ablation study on Gen SR and E2E SR across three datasets under various configurations.

| | GSM8K | | SimpleRL-Abel-Level3to5 | | DAPO-Math-17K | | Average | |
|---|---|---|---|---|---|---|---|---|
| Method | Gen SR | E2E SR | Gen SR | E2E SR | Gen SR | E2E SR | Gen SR | E2E SR |
| vanilla | 1.00 | 1.00 | 1.00 | 1.00 | 1.00 | 1.00 | 1.00 | 1.00 |
| FastGRPO | 2.66 | 2.43 | 2.91 | 2.52 | 2.78 | 2.53 | 2.78 | 2.49 |
| FastGRPO w/o online draft learning | 1.78 | 1.73 | 2.16 | 2.01 | 1.89 | 1.83 | 1.94 | 1.74 |
| vanilla w/ early termination | 1.22 | 1.21 | 1.68 | 1.61 | 1.62 | 1.58 | 1.51 | 1.47 |
| FastGRPO w/o concurrency-aware | 2.27 | 2.14 | 2.59 | 2.30 | 2.44 | 2.19 | 2.43 | 2.21 |

tion for completed sequences (denoted as vanilla w/ early termination); (2) Speculative decoding without concurrency-aware strategy and early termination (denoted as FastGRPO w/o concurrency-aware).

As shown in Table 3, the *vanilla w/ early termination* strategy provides moderate speedups, achieving generation speedup ratio values in the range of 1.2x-1.7x. However, vanilla autoregressive generation cannot effectively exploit the computational headroom that emerges as high-batch inputs reduce in concurrency due to early sequence termination. In contrast, speculative decoding inherently benefits from such underutilized capacity by enabling parallel generation. Our concurrency-aware speculative decoding approach further enhances this advantage through dynamic adjustment of decoding parameters, allowing more efficient resource utilization. The results demonstrate that our method consistently outperforms other strategies across all evaluated tasks, yielding significantly higher speedup ratios and validating the efficacy of the proposed concurrency-aware mechanism.

**The Need for Draft Model Pretraining.** We conduct an experimental analysis to evaluate the effectiveness of pretraining the draft model in accelerating the GRPO training pipeline. We design a controlled ablation scenario: the draft model is not pretrained and is trained solely through online draft learning with the target model. Speculative decoding is enabled after 128 training steps. As shown in Figure 5, although the initial token acceptance length is relatively low, it increases rapidly and converges to a level comparable to that of the pretrained draft model within just 1–2 epochs.

These results suggest that, when the sole objective is to accelerate GRPO training, pretraining the draft model may be unnecessary in most cases. The online draft learning process alone is sufficient to quickly develop a competent draft policy that enables effective speculation, reducing the need for additional pretraining overhead.

Figure 5: Average accepted length versus training steps in GRPO, with and without pretrained draft model. The model is Qwen2.5-7B-Instruct and the dataset is SimpleRL-Abel-Level3to5.

**Transfer Experiments on GRPO Variants.** To evaluate the generality of our method, we applied our acceleration framework to two GRPO variants: DAPO (Yu et al., 2025) and GPG (Chu et al., 2025). The experiments are conducted using the Qwen2.5-7B-Instruct model, with the same datasets and evaluation metrics as those used in Section 5.1. As shown in Table 3, our approach achieves significant speedups. These results indicate that our method is effective not only under standard GRPO but also in its variant frameworks, achieving over 2x end-to-end speedup ratio consistently. This demonstrates the broad applicability and robustness of our acceleration framework.

Table 4: Gen SR and E2E SR achieved by our acceleration framework on GRPO variants across multiple mathematical reasoning datasets.

| | GSM8K | | SimpleRL-Abel-Level3to5 | | DAPO-Math-17K | | Average | |
|---|---|---|---|---|---|---|---|---|
| Method | Gen SR | E2E SR | Gen SR | E2E SR | Gen SR | E2E SR | Gen SR | E2E SR |
| DAPO | 2.60 | 2.32 | 2.87 | 2.31 | 2.74 | 2.39 | 2.74 | 2.34 |
| GPG | 2.62 | 2.48 | 2.93 | 2.66 | 2.71 | 2.57 | 2.75 | 2.57 |

## 6 RELATED WORK

**Speculative Decoding.** Speculative decoding accelerates LLM inference by using a draft model to propose token sequences, which are then efficiently verified by the target model (Leviathan et al., 2023). To improve draft-target alignment, recent methods such as Eagle (Li et al., 2024b), GliDe (Du et al., 2024), and Clover-2 (Xiao et al., 2024) have explored reusing target model features or KV caches. Other efforts, including HASS (Zhang et al., 2024) and Eagle-3 (Li et al., 2025b), attempt to resolve the discrepancy between training and inference dynamics. Furthermore, SpecExec (Svirschevski et al., 2024) demonstrates the significant potential of speculative decoding for deploying models exceeding 50B parameters on consumer-grade GPUs, while APD (Israel et al., 2025) explores the decoding of diffusion large language models by adopting a parallel verification mechanism similar to speculative decoding. However, the most effective training data for a draft model is the output generated by the target model itself, yet the collection of such data is typically resource-intensive. Our framework addresses this by effectively reusing the target model's real-time generation during the RL process for joint optimization, which has not been fully explored in prior work.

**Reinforcement Learning Acceleration.** Reinforcement learning has become a key paradigm for enhancing reasoning and task-solving capabilities in large language models (LLMs) (Comanici et al., 2025; Yang et al., 2025). Methods such as GRPO (Shao et al., 2024), DAPO (Yu et al., 2025), GSPO (Zheng et al., 2025a) and others (Tan et al., 2025; Fan et al., 2025; Chen et al., 2025b;a; Zhang et al., 2025b; Chu et al., 2025; Mroueh et al., 2025; Zhao et al., 2025; Robeyns & Aitchison, 2025; Zhang et al., 2025a; Yue et al., 2025), enable stable policy optimization through clipping mechanisms. However, RL training remains highly inefficient due to long, autoregressive rollout phases that dominate end-to-end latency (Sheng et al., 2025; Zhong et al., 2025). GPU underutilization stems from both slow token generation and load imbalance caused by variable-length rollouts (Fu et al., 2025; Team et al., 2025). To mitigate these issues, recent systems such as PipelineRL (Piché et al., 2025) , StreamRL (Zhong et al., 2025) and AReal (Fu et al., 2025) decouple rollout and training via asynchronous strategies. Kimi K2 (Team et al., 2025) introduces a comprehensive RL infrastructure, optimizing across computing clusters, colocated architectures, and efficient engine switching. Furthermore, CPPO (Lin et al., 2025) and GRESO (Zheng et al., 2025b) facilitate end-to-end speedups through different clipping strategies, while TreePO (Li et al., 2025a) utilizes tree-structured sampling to improve efficiency. Despite these system-level advancements, the core bottleneck which is inefficient autoregressive rollout generation remains largely unaddressed.

## 7 CONCLUSION

In this paper, we propose a concurrency-aware speculative decoding framework with online draft learning to accelerate GRPO. Our method dynamically adjusts decoding parameters based on concurrency levels, maximizing efficiency during high-variance generation, while online joint learning maintains strong alignment between draft and target models amid parameter updates. This dual strategy sustains high token acceptance and computational efficiency throughout training. Experiments across multiple models and mathematical reasoning benchmarks show consistent end-to-end speedups of 2.35x to 2.72x, outperforming baseline approaches.

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

## A  THE OPERATIONAL INTENSITY OF ACCELERATORS

As mentioned in the main text, $I_{\text{peak}} = \frac{\text{Peak FLOPS}}{\text{Memory Bandwidth}}$. Below we calculate the $I_{\text{peak}}$ for several modern GPUs.

Table 5: The $I_{\text{peak}}$ of several modern GPUs.

| GPU | Peak BF16 Tensor Core (TFLOPS) | Memory Bandwidth (TB/s) | $I_{\text{peak}}$ |
|---|---|---|---|
| A100 40GB PCIe x 16 | 312 | 1.555 | 200.6 |
| A100 40GB SXM | 312 | 1.555 | 200.6 |
| A100 80GB PCIe x 16 | 312 | 1.935 | 161.2 |
| A100 80GB SXM | 312 | 1.555 | 153.0 |
| H100 SXM | 1979 | 3.35 | 590.7 |
| H100 PCIe | 1513 | 3.026 | 500 |
| H100 NVL | 3958 | 7.8 | 507.4 |
| H800 SXM | 1979 | 3.35 | 590.7 |
| H800 PCIe | 1513 | 2 | 756.5 |
| H200 SXM | 1979 | 4.8 | 412.3 |
| B100 | 3500 | 8 | 437.5 |
| B200 | 2250 | 8 | 562.5 |

It is obvious that $I_{\text{peak}} \gg I_{\text{GEMM}}$. This growing disparity reflects a fundamental imbalance in modern GPU architecture design: while computational throughput has scaled rapidly through specialized tensor units and algorithm-hardware co-design, memory bandwidth—constrained by physical limits—has not kept pace. As a result, the peak operational intensity ($I_{\text{peak}}$) increasingly diverges from the arithmetic intensity required by canonical operations such as GEMM (General Matrix Multiplication), indicating that most workloads will be memory-bound under naive execution.

This imbalance is not accidental, but rather a consequence of **deepening hardware constraints at the physical layer**. The bandwidth of off-chip memory interfaces, even with advanced HBM3e stacks, is approaching the limits imposed by signal integrity, power delivery, and thermal density in 2.5D/3D packaging. As NVIDIA's architecture team has noted: "We are nearing the electrical and thermodynamic boundaries of how fast we can move data across a package. Future gains must come from smarter data reuse, on-chip caching, and sparsity-aware execution, not just wider buses." (NVIDIA, GTC 2024). This explains the architectural trajectory toward higher $I_{\text{peak}}$: to sustain performance under bandwidth saturation, GPUs must extract more computation per byte transferred, effectively shifting the burden from raw bandwidth scaling to **maximizing data locality and reuse**.

Moreover, the introduction of structured sparsity (e.g., 2:4 sparsity in Ampere and beyond) and narrow-precision formats (FP8, INT4) enables GPUs like the H100, H200, and B100/B200 to operate efficiently in regimes where $I_{\text{peak}} > 500$, despite memory bandwidth improvements being sublinear relative to FLOPs growth. For instance, while the H200 doubles HBM bandwidth compared to the H100 (from 3.35 TB/s to 4.8 TB/s), its BF16 FLOPS remain unchanged at 1979 TFLOPS, resulting in a lower $I_{\text{peak}}$—a deliberate design choice to alleviate the memory bottleneck for large-model inference.

In this context, the sustained high arithmetic intensity of modern accelerators is not a sign of boundless scalability, but rather an **adaptive response to the hard limits of interconnect physics**. Future GPU generations will continue to exhibit high $I_{\text{peak}}$ not because memory bandwidth can scale indefinitely, but because they must—within the constraints of energy, area, and signal propagation—maximize computational return on every joule and every byte moved. This paradigm shift underscores the necessity of algorithm-architecture co-design in the post-Dennard scaling era, where efficiency, not just peak performance, defines the frontier of acceleration.

## B  THE MEASUREMENT METHODOLOGY OF verification_capacity

The theoretical compute-to-bandwidth ratio $I_{\text{peak}}$ is difficult to derive analytically due to hardware- and implementation-specific factors such as kernel operation, memory tiling, and SRAM constraints. Instead, we empirically estimate the effective $C_{\text{peak}}$.

---

**Algorithm 2** Empirical Measurement of $C_{\mathrm{peak}}$

---

**Require:** Target model $model$, max batch size $B_{\mathrm{max}}$, warm-up iterations $W$, repetition count $N$
**Ensure:** Estimated $C_{\mathrm{peak}}$
 1: **for** $B = 1$ to $B_{\mathrm{max}}$ **do**
 2:     Create input: $input\_ids \in \mathbb{Z}^{B \times 1}$
 3:     **for** $w = 1$ to $W$ **do**
 4:         Perform warm-up forward pass: $outputs = model(input\_ids)$
 5:     Synchronize CUDA
 6:     **for** $n = 1$ to $N$ **do**
 7:         Perform timing operation: $outputs = model(input\_ids)$
 8:         Synchronize CUDA
 9:     Append $cur\_time$ to $times$
10: Plot `times` vs. $B$
11: Detect knee/elbow point in the curve
12: **return** $B_{\mathrm{knee}}$ as $C_{\mathrm{peak}}$

---

To measure this quantity, we perform the following experiment on a given target model and accelerator (e.g., GPU): We evaluate the forward pass latency of the target model across varying batch sizes $B$, while keeping the input sequence length fixed at 1. Specifically, for each $B \in [1, B_{\mathrm{max}}]$, we feed the model a batch of $B$ identical single-token inputs (typically the EOS token), and measure the average inference time per forward pass after warm-up and multiple repetitions.

As $B$ increases, the operational intensity of the attention and feed-forward layers rises, leading to improved hardware utilization. However, beyond a certain point—when the compute demand saturates the available resources—the latency begins to grow obviously. This manifests as a clear *knee point* in the latency vs. $B$ curve (as shown in Figure 6). Algorithm 2 summarizes the measurement procedure.

It is worth noting that, since our draft model also adopts a Transformer architecture with hidden and intermediate layer dimensions identical to those of the target model, the $C_{\mathrm{peak}}$ of the draft model is the same as that of the target model. Therefore, no separate measurement is required. However, if the draft model differs in architecture or scale, its own $C_{\mathrm{draft}}$ must be measured empirically. In such cases, the optimal $K_{\mathrm{draft}}$ should satisfy $K_{\mathrm{draft}} \leq C_{\mathrm{draft}}/B$, where $B$ is the current batch size, to ensure efficient parallel verification.

## C   RELATED DATASETS

This section briefly introduces the six datasets used in our study, serving as supplementary information for the main text.

**ShareGPT-68K.**   ShareGPT-68K (Aeala, 2023) is a dialogue dataset collected from user-shared conversations on the ShareGPT platform. This cleaned split retains multi-turn human-AI interactions while removing low-quality or malformed entries, such as incomplete messages or non-text content. It provides diverse and realistic conversational samples, making it suitable for training or evaluating open-domain dialogue systems.

**GSM8K.**   GSM8K (Cobbe et al., 2021) (Grade School Math 8K) is a benchmark dataset consisting of 8,500 grade-school-level math word problems, each accompanied by a human-written, step-by-step solution. The problems require multi-step arithmetic reasoning and are designed to evaluate a model's ability to perform reliable and interpretable mathematical reasoning in natural language.

**SimpleRL-Abel-Level3to5.**   SimpleRL-Abel-Level3to5 (Zeng et al., 2025) is a curated subset of the MATH dataset, containing problems at difficulty levels 3 to 5. It is specifically designed for studying the impact of data difficulty on zero-shot reinforcement learning and model generalization. This dataset has been used in training and evaluating high-performance models such as Mistral-Small-24B and various Qwen series models.

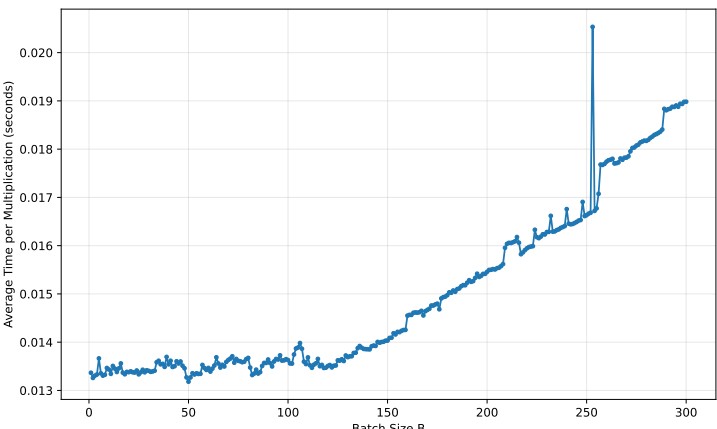

Figure 6: Forward pass latency vs. batch size $B$ for Qwen2.5-7B-Instruct on a GPU H800 SXM.

**DAPO-Math-17K.** DAPO-Math-17K (Yu et al., 2025) is a large-scale mathematical reasoning dataset comprising approximately 17,000 challenging problems across diverse domains, including algebra, calculus, and discrete mathematics. The dataset emphasizes complex reasoning and is used to assess the robustness and generalization of models in advanced mathematical problem solving, particularly under reinforcement learning settings.

**MT-Bench.** MT-Bench (Zheng et al., 2023) is a multi-turn dialogue evaluation benchmark that assesses the quality of language models in conversational settings. It features a diverse set of multi-turn scenarios covering creative writing, reasoning, and role-playing. Responses are evaluated based on coherence, consistency, and instruction following, often using LLM-as-a-judge protocols, making it a widely adopted metric for conversational capability analysis.

**HumanEval.** HumanEval (Chen, 2021) is a code generation benchmark consisting of 164 hand-written programming problems in Python. Each problem includes a function signature, docstring, and test cases. Model performance is evaluated by executing the generated code and measuring pass@1 accuracy—i.e., whether the solution passes all test cases. It is a standard benchmark for assessing functional correctness in code synthesis tasks.

## D  OTHERS

**Average Acceptance Length ($\tau$).** The average number of tokens accepted by the target model per verification step. In dynamic batch settings, $\tau$ is computed as the total number of accepted tokens divided by the total number of sequence-level verification steps. The latter is defined as the cumulative sum of the number of active (i.e., unfinished) sequences across all verification iterations. Specifically, if the batch size at iteration $t$ is $B_t$, and one verification pass is executed per active sequence, the total verification count increases by $B_t$.

