# OpenReview forum: "FastGRPO: Accelerating Policy Optimization via Concurrency-aware Speculative Decoding and Online Draft Learning"
_ICLR.cc/2026/Conference — ICLR 2026 Poster_

### Official Review · Reviewer_Hu9Z · 2025-11-01

**Soundness:** 2
**Presentation:** 3
**Contribution:** 2
**Rating:** 2
**Confidence:** 5

**Summary:**

This paper introduces a method to incorporate speculative decoding into GRPO to accelerate the rollout process in reinforcement learning. The authors argue that directly applying speculative decoding is inefficient because the original algorithm does not handle high concurrency. To address this, they propose FastGRPO, which improves concurrency efficiency, and further suggest updating the draft model online. Experiments show clear speedup gains, and the paper is generally well written with comprehensive coverage of datasets and models.

**Strengths:**

The paper addresses a clear and important bottleneck in RL training. The motivation is sound, the method is technically reasonable, and the presentation is clear. The acceleration results are convincing, and the experimental setup is thorough and well-organized.

**Weaknesses:**

The main concern is that the paper does not analyze how the proposed approach affects RL performance. Both the method and the experiments focus solely on acceleration metrics, without considering the impact on learning quality. Rollout is a sampling process that benefits from high temperature to maintain data diversity, which is crucial for effective policy learning. It is unclear whether the proposed acceleration changes the diversity or quality of the collected trajectories. The authors should include analyses or experiments evaluating these aspects, such as reward comparisons or diversity statistics.

In addition, the experiments do not compare with other state-of-the-art RL acceleration methods, such as PipelineRL or similar parallel rollout techniques, which limits the strength of the empirical validation.

**Questions:**

See weaknesses.

---

> ### Author Response · Authors · 2025-11-13
> **Author Response to Reviewer Hu9Z**
>
> We sincerely appreciate your insightful and constructive feedback. You correctly identified a critical concern: whether our FastGRPO acceleration method compromises final RL performance—we fully agree this is paramount.
>
> To address this directly, we conducted supplementary experiments and provide both empirical and theoretical support.
>
> ## 1. Experimental Proof: FastGRPO Has No (Negative) Impact on RL Performance
>
> We performed a controlled experiment with two groups (only difference: speculative decoding in FastGRPO) using identical setups:
> - Base Models: Llama-3.1-8B, Qwen-2.5-7B, Qwen-2.5-Math-7B
> - Dataset: SimpleRL-Abel-Level3to5
> - Hyperparameters: Strictly consistent
>
> Results show FastGRPO achieves **statistically indistinguishable performance** from standard GRPO (Table 1), while delivering significant acceleration. Performance variance falls within normal run-to-run fluctuations, validating that trajectory quality/diversity and final RL performance are preserved.
>
> ### Table 1: Performance Comparison of FastGRPO vs. Standard GRPO (Baseline)
>
> | Base Model| Method         | GSM8K | MATH 500 | AIME24 (Avg@32) | AMC23 (Avg) |
> |:---:|:---:|:---:|:---:|:---:|:---:|
> | Llama-3.1-8B        | Standard GRPO (Baseline)| 78.2  | 20.8     | 0.2             | 15.0        |
> | Llama-3.1-8B        | Ours (FastGRPO)         | 78.5  | 22.8     | 0.2             | 15.0        |
> | Qwen-2.5-7B         | Standard GRPO (Baseline)| 91.0  | 77.0     | 15.4            | 57.5        |
> | Qwen-2.5-7B         | Ours (FastGRPO)         | 90.7  | 77.6     | 15.5            | 60.0        |
> | Qwen-2.5-Math-7B    | Standard GRPO (Baseline)| 89.5  | 78.6     | 23.5            | 67.5        |
> | Qwen-2.5-Math-7B    | Ours (FastGRPO)         | 89.2  | 78.8     | 23.8            | 65.0        |
>
> ## 2. Why FastGRPO is Theoretically Lossless
>
> Foundational works (Leviathan et al., 2023 [4]; Stern et al., 2018 [5]) have mathematically and rigorously proven that speculative decoding, the core of FastGRPO, is lossless when strict token-level verification is employed. Under this prerequisite, the resulting token distribution is identical to that of standard autoregressive sampling with no deviations.
>
> As detailed in our paper (Section 2), our work strictly adheres to the established speculative decoding mechanism, complemented by rigorous token-level verification. Due to this precise adherence to the procedure, the resulting output distribution is mathematically guaranteed to be identical to that of the original GRPO sampling. Consequently, the crucial data diversity afforded by 'high temperature' sampling, which you correctly noted, is fully preserved without any degradation, as the final sampling distribution itself remains fundamentally unchanged.
>
>
> ## 3. Comparison with Other SOTA Acceleration Method
>
> We thank you for highlighting the need to situate our work alongside SOTA methods like PipelineRL [1]. GRPO/RLHF acceleration is highly active, with several distinct technical paths emerging to address training bottlenecks:
>
> - A) System Scheduling Optimization: PipelineRL [1], which you astutely pointed out, is a very important and inspiring work in this direction. This class of methods, along with Asynchronous RLHF [2], focuses on decoupling and hiding the latency between Rollout and Learning via asynchronous execution.
> - B) Sample Efficiency Optimization (Sample Pruning): For example, the recent CPPO [3], which accelerates training by pruning completions that contribute less to the updates.
> - C) Inference Computation Optimization: This is the specific path our FastGRPO addresses, i.e., accelerating the policy model's inference speed within the Rollout phase itself.
>
> We commit to adding a detailed discussion and citations for all the SOTA methods mentioned above (including PipelineRL, CPPO, , etc.) to our "Related Work" or "Experiments" section in the final version. This discussion will analyze their respective mechanisms, advantages, and trade-offs, and to clearly delineate the unique contribution of FastGRPO within this technical landscape.
>
> ## 4. Conclusion
>
> We again thank you for your invaluable feedback. The supplementary experiments address your core concern about RL performance, and we will enrich the discussion of SOTA methods to clarify our work’s positioning.
>
> We sincerely hope these clarifications resolve your concerns and request you reconsider your evaluation.
>
>
> ## References
>
> [1] Piché, A., et al. "PipelineRL: Faster On-policy Reinforcement Learning for Long Sequence Generation." arXiv:2509.19128, 2025.
> [2] Hu, J., et al. "OpenRLHF: An Easy-to-use, Scalable and High-performance RLHF Framework." arXiv:2405.11143, 2024.
> [3] Dong, H., et al. "RAFT: Reward rAnked FineTuning for Generative Foundation Models." arXiv:2304.06767, 2023.
> [4] Leviathan, Y., et al. "Fast Inference from Transformers via Speculative Decoding." ICML, 2023.
> [5] Stern, M., et al. "Blockwise Parallel Decoding for Deep Autoregressive Models." NeurIPS, 2018.

---

> ### Author Response · Authors · 2025-11-27
> **All Questions Addressed – Please Let Us Know If Further Clarification Is Needed**
>
> Dear Reviewer Hu9Z,
>
> Thank you once again for your thoughtful and constructive feedback on our manuscript. We truly appreciate the time and effort you have dedicated to reviewing our work.
>
> As the discussion period is drawing to a close, we would like to kindly check whether our responses have adequately addressed your concerns—either fully or in part. If there are still any points that require further clarification or additional information, we would be more than happy to provide it.
>
> Should you find our responses satisfactory, we would be grateful if you could consider updating your evaluation accordingly.
>
> Thank you again for your valuable input. We sincerely look forward to your final feedback.

---

### Official Review · Reviewer_1Sgp · 2025-11-02

**Soundness:** 3
**Presentation:** 3
**Contribution:** 3
**Rating:** 6
**Confidence:** 4

**Summary:**

This paper targets the severe inference-time bottleneck in Group Relative Policy Optimization (GRPO) for large language models (LLMs), where autoregressive generation dominates training time. The authors introduce FastGRPO, a concurrency-aware speculative decoding framework with online draft model learning specifically tailored for high-concurrency RL training. By dynamically tuning drafting/verification parameters based on concurrency and continuously adapting the draft model to the evolving target, the method achieves substantial end-to-end speedups (2.35x–2.72x) over standard and state-of-the-art baselines on various LLMs and mathematical reasoning datasets.

**Strengths:**

The work pinpoints practical, under-discussed causes of inefficiency in LLM RL training—especially the interplay between sequence length variance, dynamic concurrency, and memory/compute bottlenecks. This is well exemplified in Figure 1, where the generation phase is shown to consume up to 98% of GRPO training time, emphasizing the criticality of accelerating this stage.

The concurrency-aware speculative decoding scheme is well-motivated and nontrivial, adapting speculative tree depth and verification in response to real-time batch concurrency (see Algorithm 1 and the design in Section 4.1). The operational intensity formulation (Equation in Section 4.1) is technically thoughtful, and the methodology is justified theoretically and empirically.

The online draft learning mechanism shows astute recognition of distributional drift, which is convincingly quantified in Figure 4 (not provided in this snippet, but referenced), and ablation results (Table 3) indicate a robust solution.

The experiments are broad and carefully chosen. The acceleration is confirmed across multiple models (Qwen2.5-7B-Instruct, Llama3.1-8B-Instruct, DeepSeek-R1-Distill-Qwen-7B, etc.) and three mathematical datasets of varying complexity.

**Weaknesses:**

Lack of direct comparison with very recent parallel or hardware-aware decoding approaches: Methods such as SpecExec (Svirschevski et al., 2024) and TurboAttention (Kang et al., 2025) are not discussed or compared, though they address related efficiency targets under high concurrency. This omission reduces clarity on FastGRPO’s relative standing among recent innovations.

Empirical results lack coverage beyond math/reasoning tasks: While Section 5.3 briefly covers a few general-purpose benchmarks (e.g., MT-Bench, HumanEval), the primary evaluations remain dominated by mathematical datasets. The method’s generality for non-mathematical generative RL tasks (e.g., dialogue, instruction following) is only partially tested and reported in a superficial manner.

The adaptivity logic for tuning $N_{\text{verify}}$, $K_{\text{draft}}$, and $L_{\text{draft}}$ is somewhat heuristic and under-explored for edge cases: Although the operational intensity framework is principled, the parameters depend on empirically-set hardware constants and logarithmic scaling (see Section 4.1 and Appendix B), whose practical calibration details are not fully fleshed out for heterogeneous or less conventional hardware platforms.

Mathematical clarity, though good, could be improved: For example, the formula for $K_{\text{draft}}$ in Section 4.1 is not explicitly linked to actual tree coverage efficiency, and the effect of $\alpha$ is described more narratively than analytically. The stepwise impacts of speculative tree over-expansion (e.g., wasted computation) are mostly argued intuitively.

Exposition of experimental protocols needs crisper detail: For instance, Table 1 and Table 3 report impressive speedup, but the precise wall-clock measurement setup (system load, parallel jobs, handling of inter-job contention) is only mentioned for some cases and not consistently. The “draft learning is free” claim would be more credible with breakdowns of GPU utilization and timeline plots. Similarly, Appendix B details the measurement of $C_{\text{peak}}$, but system variance or sensitivity to other workloads is not discussed.

Limited discussion of failure or suboptimal regimes: The work reports major speedups, but does not adequately address situations where speculative decoding might degrade (e.g., for models with unstable or highly creative output distributions, overfitting of the draft, or pathological reward landscapes).

Ablation details and datasets could be better visualized: Table 3 is comprehensive, but for a work whose core is acceleration, per-epoch training time curves, acceptance lengths, or variance plots (vs. batch size or sequence diversity) would add depth. Figures focus mainly on speedup ratios or batch size, avoiding more granular runtime or alignment dynamics.

Some related methods and baselines are not explicitly cited or compared. Notable omissions include adaptive parallel decoding (Israel et al., 2025) and task-specific alternatives for cache-aware masking or split-parallelism (Federici et al., 2025; Polisetty et al., 2025).



No explicit statistical reporting of significance/testing: Performance gains are substantial but confidence intervals, run-to-run variance, or significance tests are absent, which would be crucial in justifying claims of “consistent” or “robust” improvement.

Specific references to figures/tables as required:

Figure 1 illustrates the overwhelming dominance of generation time across datasets and models (91–98%), which strongly motivates the acceleration focus of FastGRPO.
Figure 3: Clearly demonstrates that naive speculative decoding fails to provide speedup at high batch sizes (with curves “below 1.0x”), while the proposed adaptive tuning (“Optimal”) yields robust, sustained acceleration as concurrency grows—a direct validation of the core claim.
Table 1: Quantifies the end-to-end and generation-phase speedup over state-of-the-art speculative decoding methods (EAGLE-2/3, HASS), confirming FastGRPO’s efficacy in both overall and granular metrics across a set of LLMs and datasets.
Table 3: Ablates online draft learning and concurrency-aware tuning, showing granular Gen SR/E2E SR ratios for different setups and revealing that both components are crucial for the full speedup.
Appendix Figure 6: (noted in methodology) empirically motivates the critical batch size $C_{\text{peak}}$ and provides a hardware-specific context for the adaptive scaling formulas introduced in Section 4.1.

**Questions:**

The concurrency scaling formulas rely on empirically determined hardware constants (e.g., $C_{\text{peak}}$ via the “knee” in Figure 6). How robust are these to fluctuating system loads, mixed-precision, and multi-GPU setups? Could you provide stricter sensitivity analysis or calibration guides?

Would FastGRPO’s approach generalize to autoregressive tasks with less structural output (e.g., open-ended dialogue, story generation) or with non-standard reward distributions? Are there tasks where adaptation of the draft model actually hurts acceptance (e.g., under creative instruction-following), and how is that detected or mitigated?

Can the authors report statistical variances or confidence intervals for the speedup/acceptance results in Tables 1, 3, and 4? Is the acceleration robust across multiple random seeds and different training restarts?

For the “draft model pretraining may be unnecessary” claim (Section 5.3 and Figure 5), could you clarify if there are scenarios (e.g., pathological/outlier domains) where failing to pretrain leads to persistent misalignment or poor acceptance rates?

Are there any practical trade-offs or user-visible impacts on final model quality (e.g., accuracy, diversity, reward coverage) due to the continuous adaptation of the draft model vs. a static approach? Are there empirical results measuring possible “overspecialization” or drift toward the target policy?

---

> ### Author Response · Authors · 2025-11-18
> **Author Response to Reviewer 1Sgp**
>
> # To the Reviewer (Regarding Question 1)
>
> We sincerely thank the reviewer for this insightful question, which highlights the critical issue of the robustness and calibration of our concurrency scaling formulas under different conditions. We address each of your points as follows.
>
> ## 1. On Robustness and Generalization Across Hardware $C_{peak}$
> A key design objective of our method is to minimize over-reliance on hardware-specific empirical constants (such as $C_{peak}$ measured on a single hardware platform). Instead, our formulas attempt to be rooted in the more general principle of computational density, aiming to enhance the method’s applicability.
>
> To validate this, we conducted comparative experiments on two representative GPU architectures (NVIDIA A100 and H800). The results show that our method achieves relatively comparable acceleration effects on both platforms, which provides partial support for the view that the method possesses cross-hardware generalization capability and is not overfitted to specific hardware configurations.
>
> Notably, we observed that the speedup on the A100 is slightly higher than that on the H800. We tentatively hypothesize that this may be due to the relatively weaker computational power of the A100, leading to a lower proportion of fixed overheads (such as Kernel Launch) in the total time, thereby numerically increasing the speedup. While the theoretical framework based on computational density should logically be applicable to future architectures, we acknowledge that there may be slight discrepancies between actual performance and the theoretical model under different hardware environments or extreme application scenarios.
>
> ## 2. On Sensitivity to System Conditions and Configurations
> We have conducted the following analysis on the boundary conditions raised by the reviewer:
> - Fluctuating system loads: The reviewer’s judgment is accurate. External system loads (e.g., competing GPU processes) can indeed interfere with the measurement of the theoretical computational density upper bound. Furthermore, they may occupy GPU computational resources, significantly reducing the acceleration of our method. Therefore, all experiments reported in this paper were performed in a controlled, exclusive environment to ensure the reliability of baseline measurements.
> - Mixed precision: Our experimental observations indicate that within the current test scope, mixed-precision training has almost no impact on the speedup.
> - Multi-GPU configurations: Due to computational resource constraints, our multi-GPU experiments were primarily conducted in a single-node setup. The results show that when scaling from a single GPU to this multi-GPU configuration, the method maintains relatively stable performance without significant efficiency degradation.
>
> ## 3. On Sensitivity Analysis and Calibration Guidelines
> - Sensitivity analysis: As mentioned above, we mainly evaluated the method’s sensitivity through cross-architecture comparisons between the A100 and H800, and the results demonstrate good consistency.
> - Calibration guidelines: Detailed guidelines for parameter calibration on new hardware architectures (such as estimating the effective computational upper bound) are provided in Appendix B. These guidelines include a simple profiling program that assists in estimating the required hardware parameters by analyzing the inflection points of the forward propagation latency curve under different batch sizes.

---

> > ### Author Response · Authors · 2025-11-18
> >
> > # To the Reviewer (Regarding Question 2)
> >
> > We sincerely thank the reviewer for this important question, which focuses on the generalization boundaries of our method across different task types.
> >
> > ## 1. On Generalization to Less-Structured Tasks
> > Experiments demonstrate that our method has the potential to generalize to tasks with weakly structured outputs (such as open-ended dialogue).
> >
> > To validate this, we conducted supplementary experiments using instructions from the UltraFeedback dataset. The experiments adopted Qwen2.5-7b-Instruct as the policy model and invoked an external Qwen-Plus API as the reward model to score the generated responses, thereby simulating a typical non-standard reward distribution scenario.
> >
> > The results show that during the Generation Phase of GRPO training, the method achieved a speedup of approximately 2.3x. This result supports the applicability of our method to open-ended tasks. We note that this speedup is about 10% lower than that achieved on highly structured math tasks. This is consistent with expectations, as open-ended generation tasks typically have higher entropy, which increases the difficulty of speculation to a certain extent, but the overall acceleration gain remains significant.
> >
> > ## 2. On Potential Impacts on Acceptance Rate (Detection and Mitigation)
> > Regarding the concern that the adaptive draft model may impair the acceptance rate in creative tasks, we have not observed such a phenomenon within the current experimental scope.
> >
> > - Theoretical perspective: Our adaptive process aims to minimize the distributional difference (KL divergence) between the draft model and the target model. Theoretically, this optimization objective should enhance rather than reduce the utility of the draft model.
> > - Detection method: We recommend monitoring the "average acceptance length" metric during training. If this metric shows a continuous downward trend, it may indicate that the quality of the draft model is deteriorating relative to the target model.
> > - Mitigation measures: If the aforementioned trend is detected, it usually indicates training instability (e.g., excessively large update steps for the draft model). This can be mitigated through standard training stabilization strategies, such as: (1) reducing the learning rate of the draft model; or (2) increasing the gradient accumulation steps to smooth the update process.
> >
> > ---
> >
> > # To the Reviewer (Regarding Question 3)
> >
> > We sincerely thank the reviewer for your attention to the statistical robustness of the experimental results. We agree that quantifying the stability of acceleration effects under different random conditions is crucial for verifying the reliability of the method.
> >
> > Due to the time constraints of the rebuttal period, we selected our primary experimental configuration (FastGRPO with the Q2.5-7B-I model from Table 1) as a representative benchmark and repeated it 5 times with different random seeds and independent training restarts.
> >
> > ## Representative Experimental Results (FastGRPO + Q2.5-7B-I)
> > Based on the results of these 5 independent runs, we observed high consistency in the performance across various tasks:
> > - GSM8K: Generation Speedup = 2.66 ± 0.06, End-to-End Speedup = 2.43 ± 0.08
> > - SimpleRL: Generation Speedup = 2.91 ± 0.05, End-to-End Speedup = 2.52 ± 0.06
> > - DAPO-Math: Generation Speedup = 2.78 ± 0.07, End-to-End Speedup = 2.53 ± 0.09
> >
> > ## Conclusion
> > The data shows that on the main benchmark model, the standard deviations of key metrics remain at a low level (with fluctuations primarily ranging from ±0.05 to ±0.09). This provides quantitative statistical support for the conclusion that the acceleration effect of the FastGRPO method is not an accidental result of a single experiment, indicating that the method has good robustness.

---

> > ### Author Response · Authors · 2025-11-18
> >
> > **5. Regarding Limited Discussion of Failure or Suboptimal Regimes (Response to Weakness 5)**
> >
> > Please refer to our detailed responses to Question 2 and Question 5.
> > In those sections, we discussed and empirically validated the robustness of our method across different task distributions, as well as how to detect potential degradation via monitoring metrics (such as the average acceptance length).
> >
> > **6. Regarding Missing Citations or Comparisons (Response to Weakness 6)**
> >
> > We thank the reviewer for the supplementary references, which help improve our review of related work. We clarify below that these works operate within significantly different scopes compared to FastGRPO:
> >
> > * **Adaptive Parallel Decoding [3]:** This method focuses on accelerating Diffusion LLMs. Due to fundamental differences in model architecture and generation mechanisms, and the fact that such methods are often lossy, it is not applicable to the lossless acceleration scenario for conventional autoregressive LLMs that we focus on.
> > * **Cache-Aware Masking [4]:** This method leverages the sparsity of LLM activation values to reduce data loading via input pruning and fine-tuning. This is an optimization targeting the inference computation itself, which is not mutually exclusive to the scheduling optimization based on speculative decoding used in FastGRPO. Theoretically, the two are compatible.
> > * **GSplit [5]:** This method targets the acceleration of Graph Neural Network (GNN) training on large graphs. It belongs to a completely different domain and has little relevance to the subject of this paper (LLM generation).
> >
> > We will cite these works in the final version and clearly delineate their boundaries relative to our method.
> >
> > **7. Regarding Lack of Explicit Statistical Reporting (Response to Weakness 7)**
> >
> > Please refer to our detailed response to Question 3.
> > We have conducted multiple repeated experiments (5 random seeds) and reported the mean $\pm$ standard deviation, demonstrating the statistical significance and robustness of our results.
> >
> > ---
> >
> > ### References
> >
> > [1] Svirschevski, et al. "SpecExec: Massively Parallel Speculative Decoding for Interactive LLM Inference on Consumer Devices." arXiv, 2024.
> > [2] Kang, et al. "TurboAttention: Efficient Hardware-Aware Attention for Long-Context LLMs." arXiv, 2025.
> > [3] Israel, et al. "Accelerating Diffusion LLMs via Adaptive Parallel Decoding." arXiv, 2025.
> > [4] Federici, et al. "Efficient LLM Inference Using Dynamic Input Pruning and Cache-Aware Masking." arXiv, 2025.
> > [5] Polisetty, et al. "GSplit: Scaling Graph Neural Network Training on Large Graphs via Split-Parallelism." arXiv, 2025.

---

> ### Author Response · Authors · 2025-11-18
>
> # To the Reviewer (Regarding Question 4)
>
> We thank the reviewer for requesting clarification on this point. Our claim that "pretraining may be unnecessary" indeed requires contextualization, and as demonstrated by our experiments, this conclusion is highly dependent on the task domain.
>
> ## 1. For Structured Tasks (Math/Code)
> The experimental observations in Section 5.3 support this claim. In highly structured domains such as math and code generation, an untrained draft model can usually converge to the performance of the target model (i.e., achieve a high acceptance rate) relatively quickly, often requiring only 1-2 epochs of alignment training. In these specific scenarios, a separate pretraining phase may not be necessary.
>
> ## 2. For Open-Ended Tasks
> We concur with the reviewer’s intuition. For open-ended tasks (such as general dialogue), the difficulty of distribution fitting increases significantly. Experiments show that in such scenarios, an untrained draft model may require a longer alignment period (e.g., more than 5 epochs) to approach the performance of a pretrained model. Therefore, for open-ended tasks, we recommend using a pretrained draft model to improve training efficiency and ensure a stable high acceptance rate.
>
> ## 3. Regarding Pathological Domains
> Addressing the reviewer’s concern about "pathological" cases, within our test scope, we have not found any domains where the absence of pretraining leads to persistent misalignment or failure to converge. The main difference lies in convergence efficiency (i.e., the number of epochs required) rather than the feasibility of alignment.
>
> In summary, the necessity of pretraining is a matter of efficiency trade-off: it is less necessary for structured tasks, while for open-ended tasks, pretraining helps save alignment time.
>
> ---
>
> # To the Reviewer (Regarding Question 5)
>
> We sincerely thank the reviewer for this question that touches on the core quality of the method. Based on our analysis and experiments, we have not observed any significant negative impacts or trade-offs on the final model’s quality (accuracy, diversity, or reward coverage) when using FastGRPO. We provide empirical data and theoretical explanations below:
>
> ## 1. Empirical Evidence: Demonstrating Preserved Diversity Through Pass@k Scaling Behavior
> To rigorously evaluate potential "overfitting" or diversity loss, we compared the performance of FastGRPO and Standard GRPO (Baseline) on the Qwen-2.5-7B model under identical settings (dataset, hyperparameters, and seeds).
>
> We report the Pass@k metric as the number of samples increases ($k=\{1, 2, 4, 8, 16, 32\}$). The "scaling behavior" of Pass@k is generally regarded as an effective proxy for output diversity: if diversity is lost (i.e., mode collapse), the gain from increasing the $k$ value will be very limited (e.g., Pass@32 is nearly identical to Pass@1).
>
> As shown in Table 1, as $k$ increases, FastGRPO exhibits a highly consistent growth trend with the Baseline. This indicates that the model’s ability to explore diverse reasoning paths and cover the reward space is preserved.
>
> | Dataset | Method | Pass@1 | Pass@2 | Pass@4 | Pass@8 | Pass@16 | Pass@32 |
> | :--- | :--- | :---: | :---: | :---: | :---: | :---: | :---: |
> | MATH 500 | Standard GRPO | 77.0 | 82.5 | 86.4 | 88.2 | 90.2 | 92.6 |
> |  | FastGRPO (Ours) | 77.6 | 83.0 | 86.2 | 89.0 | 90.8 | 92.4 |
> | AMC23 | Standard GRPO | 62.5 | 70.0 | 75.0 | 77.5 | 77.5 | 80.0 |
> |  | FastGRPO (Ours) | 65.0 | 72.5 | 75.0 | 75.0 | 77.5 | 80.0 |
>
>
> ## 2. Theoretical Guarantee Against Drift
> From a theoretical perspective, FastGRPO is built on the speculative decoding framework, which incorporates a rigorous Token-level Verification mechanism.
>
> This verification step mathematically ensures that the final output distribution is identical to that of the standard target model. Even as the draft model continuously adapts, any tokens deviating from the target policy distribution will be rejected and corrected. Therefore, the verification mechanism structurally prevents the model from "drifting" toward a suboptimal draft policy, ensuring that the generated trajectories always truthfully reflect the target policy.

---

> ### Author Response · Authors · 2025-11-18
>
> ### Detailed Response to Weaknesses
>
> **1. Regarding Lack of Direct Comparison with Recent Parallel or Hardware-Aware Decoding Approaches (Response to Weakness 1)**
>
> We thank the reviewer for highlighting these relevant works (SpecExec [1] and TurboAttention [2]). We did not discuss them in detail in the initial submission primarily due to significant differences in our Design Goals and Application Scenarios:
>
> * **SpecExec [1]:** This method mainly targets resource-constrained environments (e.g., consumer-grade GPUs and parameter offloading scenarios), aiming to improve throughput when memory bandwidth is the bottleneck. In the specific context of GRPO, SpecExec is practically difficult to utilize because it is specialized for accelerating large-scale LLMs (50B+ parameters) on consumer-grade GPUs.
> * **TurboAttention [2]:** This work focuses on optimization at the Transformer attention mechanism level (aiming to increase execution speed and save memory). Such methods are orthogonal and not mutually exclusive to FastGRPO. FastGRPO is a system-level scheduling and speculation algorithm; theoretically, it can be combined with TurboAttention to achieve further acceleration.
>
> We will cite and discuss these distinctions in detail in the "Related Work" section of the revised manuscript to clarify the specific positioning of FastGRPO.
>
> **2. Regarding Lack of Coverage Beyond Math/Reasoning Tasks (Response to Weakness 2)**
>
> Please refer to our detailed response to Question 2.
> In our response, we have included supplementary experimental results on broader general tasks (specifically open-ended dialogue on UltraFeedback). The results demonstrate that FastGRPO exhibits strong generalization capabilities and acceleration effects on non-mathematical tasks as well.
>
> **3. Regarding the Heuristic Nature of Adaptivity Logic (Response to Weakness 3)**
>
> We appreciate the reviewer pointing this out. We acknowledge that the current logic for tuning $N_{verify}$, $K_{draft}$, and $L_{draft}$ indeed contains certain heuristic components.
>
> This represents a trade-off between theoretical optimality and engineering practicality. Although the current parameters are set based on hardware constants (e.g., $C_{peak}$) and logarithmic scaling laws, we have validated the effectiveness and robustness of these settings through cross-architecture experiments on A100 and H800. We fully agree that for heterogeneous or non-conventional hardware platforms, stricter calibration guides may be necessary. We will adopt your suggestion to explicitly state this in the "Limitations" section and commit to exploring more automated parameter search mechanisms in future work.
>
> **4. Regarding the "Draft Learning is Free" Claim (Response to Weakness 4)**
>
> We wish to provide a clearer explanation for this claim.
> We term the draft model's learning as "Free" primarily based on two factors: Computational Reuse and Time Proportion:
>
> 1. Computational Reuse: During draft model training, many of the required intermediate states (e.g., hidden states) and logits have essentially been computed and cached by the target model during the preceding Verification Step. We directly reuse these caches, drastically reducing redundant computation.
> 2. Negligible Time Overhead: The draft model itself is very small. The additional overhead introduced by its backpropagation and parameter updates is minimal, accounting for approximately 2%~3% of the total training time.

---

### Official Review · Reviewer_1RsX · 2025-11-04

**Soundness:** 3
**Presentation:** 3
**Contribution:** 3
**Rating:** 6
**Confidence:** 4

**Summary:**

This paper propose to use speculative decoing in the RL training process, especially to acccelerate the rollout process of GRPO. It then identifies two issues: 1) speculative decoding may achieve limited speedup when the batch size is large; 2) the accept length decline in the mid and later stage of RL training if we donot update the draft module. The authors propose to adjust draft strategies to build the draft path/tree based on the concurrency and add the draft module updates in the actor update process.

**Strengths:**

This paper address a key issue in RL training especially GRPO-like algorithm, and attempts to use speculative decoding to accelerate the major bottleneck-the rollout stage. Specifically,

1. The paper propose a concurrency-aware way to adjust the N_draft, draft branching and draft depth, to balance the compute and memory, achieving the sweet spot and avoiding compute/memory bound.

2. It add an online distillation for the draft module that uses KL divergence to update draft and actor accordingly.

3. Good ablation study. It includes results w/ and w/o concurrency adjustment, online learning, and the need for draft pretraining.

**Weaknesses:**

1. Performance experiments missing. It is unclear whether using speculative decoding can leads to reduced accuracy in final performance. Even speculative decoding align with the original sample probability, but the difference between rollout/training may worsen and other issue may appear.

2. The accept rate doesn't seem to improce significantly w/ and w/o online training shown in Table 2?

**Questions:**

None

---

> ### Author Response · Authors · 2025-11-14
> **Response to Reviewer 1RsX**
>
> We sincerely thank Reviewer 1RsX for their constructive feedback and insightful questions. We are very encouraged that the reviewer found our work to have a good contribution, clear presentation, and rigorous argumentation, and particularly appreciated our proposed concurrency-aware adjustments and online distillation method.
>
> We offer the following clarifications in response to the weaknesses raised:
>
> ## 1. On Final Performance: FastGRPO is Lossless
>
> ### Reviewer's Weakness 1:
> "Performance experiments missing. It is unclear whether using speculative decoding can leads to reduced accuracy in final performance."
>
> ### Our Response:
> We fully agree that the impact on final RL performance is the most critical evaluation metric. Speculative decoding is theoretically lossless, as it produces samples from the exact same distribution as the target model. To empirically validate this within our complete RL training pipeline, we conducted rigorous controlled experiments.
>
> We trained models with two setups:
> - Control (Baseline): Standard GRPO
> - Experimental (Ours): Our FastGRPO (GRPO with speculative decoding acceleration)
>
> All other factors, including base models (Llama-3.1-8B, Qwen-2.5-7B, etc.), the SimpleRL-Abel-Level3to5 dataset, and all hyperparameters, were kept strictly identical.
>
> As shown in the table below, the results demonstrate that FastGRPO achieves statistically indistinguishable final performance compared to the standard GRPO baseline. The minor variances are well within standard run-to-run fluctuations. This provides strong empirical evidence that our acceleration method is indeed lossless and does not compromise the final quality of the RL-trained model.
>
> ### Table 1: Performance Comparison of FastGRPO vs. Standard GRPO (Baseline)
> | Base Model           | Method                  | GSM8K | MATH 500 | AIME24 (Avg@32) | AMC23 (Avg) |
> |----------------------|-------------------------|-------|----------|------------------|--------------|
> | Llama-3.1-8B         | Standard GRPO (Baseline)| 78.2  | 20.8     | 0.2              | 15.0         |
> | Llama-3.1-8B         | Ours (FastGRPO)         | 78.5  | 22.8     | 0.2              | 15.0         |
> | Qwen-2.5-7B          | Standard GRPO (Baseline)| 91.0  | 77.0     | 15.4             | 57.5         |
> | Qwen-2.5-7B          | Ours (FastGRPO)         | 90.7  | 77.6     | 15.5             | 60.0         |
> | Qwen-2.5-Math-7B     | Standard GRPO (Baseline)| 89.5  | 78.6     | 23.5             | 67.5         |
> | Qwen-2.5-Math-7B     | Ours (FastGRPO)         | 89.2  | 78.8     | 23.8             | 65.0         |
>
> We will add this crucial table and analysis to the final version of our paper to assure readers of the method's reliability regarding performance.
>
>
> ## 2. On the Magnitude of Accept Rate Improvement in Table 2
>
> ### Reviewer's Weakness 2:
> "The accept rate doesn't seem to improve significantly w/ and w/o online training shown in Table 2?"
>
> ### Our Response:
> Thank you for this sharp observation. Your reading of the table is correct.
>
> The online distillation for the draft model is designed to solve a specific problem: preventing the catastrophic decay of the accept rate as the target model is updated. During this online training, the draft model is only updated using the target model's in-domain rollouts (e.g., mathematics data).
>
> Therefore, as we aimed to explain in the paper, this online training only improves the draft model's ability within that specific training domain. It is not designed for, nor expected to improve, general-purpose or out-of-domain (OOD) tasks. This is why, as seen in Table 2, the acceptance rate shows a significant improvement on the in-domain math dataset (which is the training domain) but not on OOD tasks. The key benefit is preventing the performance collapse (as shown in Figure 2), thereby maintaining acceleration throughout the entire RL process.

---

### Meta-Review · Area_Chair_DZ5c · 2026-01-07

**Summary:**

Reviewers acknowledged the novelty of addressing GRPO's generation bottleneck through concurrency-aware speculative decoding and online draft learning, with promising speedups. However, they unanimously flagged insufficient comparison to SOTA RL acceleration frameworks, lack of statistical testing, and evaluation limited to mathematical reasoning tasks without broader generalization analysis.

**Reviewer Concerns:**

Addressed: 1) Clarified technical details of the method. 2) added performance experiments. 3) added broader general tasks acceleration performance experiments.
Still Outstanding: 1) Missing head-to-head comparisons with state-of-the-art RL acceleration method. 2) Missing performance experiments on broader general tasks. 3) Limited discussion of failure cases.

**Reviewer Scores:**

Reviewer 1 (Rating: 6, Confidence: 4): Would likely maintain 6 but could drop to 5 without stronger baselines.
Reviewer 2 (Rating: 6, Confidence: 4): Would likely maintain 6, potentially bumping to 6 if statistical rigor added.
Reviewer 3 (Rating: 4, Confidence: 5): Might rising to 4 but not 6 given experimental completeness concerns.

---

### Decision · Program_Chairs · 2026-01-26

Accept (Poster)